# Auto-Detection of Motion Artifacts on CT Pulmonary Angiograms with a Physician-Trained AI Algorithm

**DOI:** 10.3390/diagnostics13040778

**Published:** 2023-02-18

**Authors:** Giridhar Dasegowda, Bernardo C. Bizzo, Parisa Kaviani, Lina Karout, Shadi Ebrahimian, Subba R. Digumarthy, Nir Neumark, James M. Hillis, Mannudeep K. Kalra, Keith J. Dreyer

**Affiliations:** 1Massachusetts General Hospital, Harvard Medical School, Boston, MA 02114, USA; 2Mass General Brigham Data Science Office, Boston, MA 02114, USA

**Keywords:** artificial intelligence, motion artifact, CT pulmonary angiography, quality improvement

## Abstract

**Purpose**: Motion-impaired CT images can result in limited or suboptimal diagnostic interpretation (with missed or miscalled lesions) and patient recall. We trained and tested an artificial intelligence (AI) model for identifying substantial motion artifacts on CT pulmonary angiography (CTPA) that have a negative impact on diagnostic interpretation. **Methods**: With IRB approval and HIPAA compliance, we queried our multicenter radiology report database (mPower, Nuance) for CTPA reports between July 2015 and March 2022 for the following terms: “motion artifacts”, “respiratory motion”, “technically inadequate”, and “suboptimal” or “limited exam”. All CTPA reports were from two quaternary (Site A, *n* = 335; B, *n* = 259) and a community (C, *n* = 199) healthcare sites. A thoracic radiologist reviewed CT images of all positive hits for motion artifacts (present or absent) and their severity (no diagnostic effect or major diagnostic impairment). Coronal multiplanar images from 793 CTPA exams were de-identified and exported offline into an AI model building prototype (Cognex Vision Pro, Cognex Corporation) to train an AI model to perform two-class classification (“motion” or “no motion”) with data from the three sites (70% training dataset, *n* = 554; 30% validation dataset, *n* = 239). Separately, data from Site A and Site C were used for training and validating; testing was performed on the Site B CTPA exams. A five-fold repeated cross-validation was performed to evaluate the model performance with accuracy and receiver operating characteristics analysis (ROC). **Results**: Among the CTPA images from 793 patients (mean age 63 ± 17 years; 391 males, 402 females), 372 had no motion artifacts, and 421 had substantial motion artifacts. The statistics for the average performance of the AI model after five-fold repeated cross-validation for the two-class classification included 94% sensitivity, 91% specificity, 93% accuracy, and 0.93 area under the ROC curve (AUC: 95% CI 0.89–0.97). **Conclusion**: The AI model used in this study can successfully identify CTPA exams with diagnostic interpretation limiting motion artifacts in multicenter training and test datasets. **Clinical relevance**: The AI model used in the study can help alert technologists about the presence of substantial motion artifacts on CTPA, where a repeat image acquisition can help salvage diagnostic information.

## 1. Introduction

The chest represents one of the most frequently scanned body parts in CT, but also ranks highly among the most challenging parts to obtain optimal diagnostic quality. Recent technical advancements in multidetector-row CT scanners have led to tremendous improvements in diagnostic quality with lower noise, higher contrast, and fewer motion artifacts. Although with faster scanning times and better reconstruction techniques, most chest CT exams are generally optimal for diagnostic interpretation, several artifacts can still have a negative impact on diagnostic interpretations of chest CT [1]. The common artifacts include beam hardening, photon starvation, partial volume, metal, motion, and cone-beam artifacts [2,3]. Such artifacts can hinder the optimal evaluation of lung abnormalities as well as mediastinal and vascular findings. 

Prior studies have reported that motion artifacts are frequent, especially on legacy scanners, and can limit the diagnostic information from routine chest CT and CT pulmonary angiography (CTPA). Motion artifacts can result in misinterpretation as the artifacts can mimic an embolus or the artifact can cause an apparent abrupt vessel cut-off [4,5]. In the lung parenchyma, artifacts can reduce the ability to detect and characterize both focal (such as lung nodules) and diffuse parenchymal processes. Such artifacts are especially common in CTPA of critically ill patients and patients with shortness of breath and/or persistent cough [6]. The use of wide-area detector scanners and scan capabilities such as high non-overlapping pitch and faster rotation time reduce scanning duration and have lower artifacts [3,7]. Such fast acquisition modes are incompatible with dual-energy CT and patients with large body habitus. Furthermore, such scanners and advanced techniques still represent a minority of clinically deployed scanners, even in developed countries such as the United States [8]. 

Prior studies have reported the utility of AI in identifying motion artifacts. However, the studies have either been conducted on CT coronary angiography or involve AI segmentation of regions with motion artifacts as opposed to the detection of motion artifacts on non-ECG-gated CTPA [9,10,11]. Furthermore, the AI model was trained by physicians without programming knowledge or data science background. An automated method of detecting substantial motion artifacts at the time of scanning can help rescan patients during the same imaging session with better coaching or faster scanning techniques. Despite technological advances, our prior study reported a 35% prevalence rate of motion artifacts on chest CT [12]. A vendor-agnostic motion artifact detection model can thus extend the applications of such models to chest CT from the current vendor-specific applications in coronary CT angiography. User-trained AI models such as ours can empower non-computer programmers to create local solutions to address local issues. The local use of such models would not require the onerous burden of FDA clearance, thus facilitating their immediate use to target issues. Recent advances in machine and deep learning (DL) have led to the creation of several AI algorithms in medical imaging, including image reconstruction, triaging, quality control, and pathology detection [13,14,15,16,17,18]. Therefore, we trained and tested an artificial intelligence (AI) model to identify substantial motion artifacts on CTPA that have a negative impact on diagnostic interpretation.

## 2. Materials and Methods

### 2.1. Study Design

Our retrospective study was conducted after receiving approval from the Institutional Review Board (IRB). The study was Human Insurance Portability and Accountability Act (HIPAA)-compliant. The study methodology has been described according to CLAIM guidelines [19]. The model was trained and tested by physicians without prior knowledge of or training in machine learning or coding. 

### 2.2. Data Definitions

The study included adult patients (≥19 years) who underwent CTPA at one of the three hospitals (quaternary hospitals: Massachusetts General Hospital, Brigham and Women’s Hospital; community hospital: Cooley Dickinson Hospital) within an integrated health system. A commercial radiology report search engine, Nuance mPower, was used to identify radiology reports with a mention of motion artifacts in CTPA examinations performed between January 2015 and November 2021. The following keywords were used to identify the eligible CTPA: “motion artifacts”, “respiratory motion”, “technically inadequate”, and “suboptimal” or “limited exam”. The search was optimized to identify positive CTPA with the keyword hits (without mention of “absence” or “no” within a few words of the keywords). A negative search with the exact keywords was used to identify the control CTPA exams without motion artifacts. 

All CTPA exams were performed on one of the 28 CT scanners in the three participating sites with single- or dual-energy CT protocols using the standard of care scan protocols. For each CTPA, a thin slice coronal multiplanar image at the level of descending thoracic aorta was de-identified and exported offline from the PACS workstation. 

### 2.3. Ground Truth

In addition to the radiology reports, a thoracic radiologist (MKK with 16 years of subspecialty experience) reviewed all CTPA and opined on the presence or absence of substantial motion artifacts. Each CTPA thus had the opinions of two radiologists: the reporting radiologist and the study coinvestigator radiologist. Substantial motion artifacts were defined as CTPA exams with the presence of motion artifacts involving both lungs (at least 50% of each lung) and limiting the ability to assess pulmonary embolism or parenchyma. Minor motion artifacts involving a single lobe or smaller portions of the lungs without effect on the diagnostic evaluation of pulmonary embolism or parenchyma were labeled as negative for substantial motion artifacts. Although extremely common, minor artifacts are not as important since they should not trigger repeat imaging. CTPA with evidence of “white lungs” (diffuse parenchymal opacities), substantial bilateral lung volume loss, or pneumonectomies were excluded from the study (*n* = 50 CTPA exams). These cases were excluded because it is difficult to assess motion artifacts’ impact on the pulmonary evaluation in such cases. CTPA exams without the complete inclusion of lung apex and bases were excluded from the study.

### 2.4. Model

The AI model was trained on a deep learning model-building platform, Vision Pro Deep Learning (VPDL, COGNEX Corporation, Natick, MA, USA). The software enables users to train DL models based on a labeled image dataset using a vision-optimized deep neural network. The users require no formal programming or coding knowledge or experience. The VPDL platform automatically pre-processes the images for training, such as image augmentation, setting class weight, or oversampling of the imbalanced classes. The users have the provision to re-size the images, set the threshold, and perturbate with rotation, flip, zoom, and blur if required. The users can upload the images directly onto the graphic user interface. The VPDL platform has two different options for training classification models: High Detail Mode (HDM) and Focused Mode (FM). The HDM enables model training for challenging or complex applications (such as pixel-level information in the image domain) and provides higher accuracy. A heat map is also generated in HDM that indicates the image region that was most influential in the classification decision (explainable AI) [20,21]. FM enables the fast training of models for simple applications (such as the distinction between different image types or body parts). We used the HDM classification model for identifying CTPA examinations with substantial motion artifacts. Further details on the model architecture can be found in a prior study [22]. The model architecture and training workflow is represented in Figure 1. 

### 2.5. Training

A physician co-investigator (GD with one-year post-doctoral research fellowship experience in thoracic imaging) trained the AI model on the VPDL platform without prior programming or data science knowledge. The images were de-identified, exported from the PACS workstation (Visage), and then uploaded onto the software platform installed on a virtual machine within the hospital intranet to maintain data security and privacy. Within the platform, the study coinvestigator labeled each uploaded image as “motion” (with substantial motion artifacts) or “no motion” (without substantial motion artifacts) based on the assessment from the thoracic radiologist. In the first training, all CTPA examinations from three hospitals were included. The software randomized the training and validation dataset with a 70% and 30% distribution respectively. We performed five-fold repeated cross-validation to evaluate the robustness of the model. The output was recorded and separately analyzed.

To establish the inter-institutional generalizability of the model, we trained a model using images from two hospitals (A and C, after excluding the data from Site B) and then tested the algorithm on CTPA data from the third hospital (Site B). 

### 2.6. Statistical Analysis

Information on the distribution of true positive, true negative, false positive, and false negative CTPA was recorded in Microsoft Excel worksheets (Microsoft Inc., Redmond, WA, USA). The data were analyzed with SPSS statistical software, version 26 (IBM Inc., Armonk, NY, USA). The performance of the AI model was evaluated using sensitivity, specificity, accuracy, and area under the curve (AUC) for the receiver operating characteristic (ROC) analysis. The average of the five models was considered for the five-fold repeated cross-validation. We also estimated the F-score for AI model performance to measure the harmonic average of precision and recall/sensitivity.

## 3. Results

Our study included 793 CTPA examinations from 793 adult patients (mean age = 63 ± 17; 391 men, 402 women). The distribution of CTPA across each site is as follows: Site A, *n* = 335; Site B, *n* = 259; Site C, *n* = 199. A total of 455/793 (57%) CTPA examinations were performed during the emergency visit, while 277/793 (35%) and 111/793 (14%) examinations were among inpatients and outpatients, respectively. Most CTPA with substantial motion artifacts were either from the emergency department (*n* = 213/455) or inpatients (*n* = 146/227), with a minority of patients coming with outpatient referrals (*n* = 33/111). There was no discrepancy between the reporting radiologists and the research radiologist for the presence of substantial motion artifacts.

### 3.1. Model Validation at Three Sites

Among CTPA exam datasets from all three sites in model training, 471 CTPA (*n* = 421/793, 53.1%) had substantial motion artifacts, and 372 CTPA (*n* = 372/793, 46.9%) were without substantial artifacts. For the average performance after five-fold repeated cross-validation for the two-class classification (as either with or without substantial motion artifacts), on the 30% validation dataset, the AI model had a sensitivity of 94%, specificity of 91%, 93% accurate with AUC of 0.93 (95% confidence interval (95% CI) 0.89–0.97). The best-performing model had F-scores of 96% and 95% for identifying CTPA with and without substantial motion artifacts, respectively. The two-class classification of the AI model is summarized in Figure 2.

### 3.2. Model Testing

External testing with Sites A and C training datasets and Site B as the test site, the model performance statistics were 85% sensitivity, 90% specificity, 86% accuracy, and an AUC of 0.87 (95% CI 0.82–0.92). One hundred and forty-seven CTPA exams were correctly classified into those with substantial motion artifacts (true positive), and nine CTPA were mislabeled as positive for motion artifacts (false positive). Seventy-seven CTPA exams were correctly classified as without motion artifacts (true negative), and twenty-six CTPA exams were misclassified as without motion (false negative). The confusion matrix and AUC of model performance is represented in Figure 3 and Figure 4 respectively, and the model performance for validation and test data is summarized in Table 1.

## 4. Discussion

We reported high accuracy, sensitivity, and specificity of physician-trained and tested AI models for identifying substantial motion artifacts on CTPA examinations. Prior studies have reported on the ability of AI algorithms to identify anatomic regions with motion artifacts. Our study uses a single coronal multiplanar reformatted image per CT exam and, therefore, could be more time efficient while ignoring region-specific, sparse motion artifacts that do not require repeat acquisition [9,10,22]. To our best knowledge, there are no peer-reviewed reports on using AI models to identify motion artifacts in CTPA or chest CT examinations. 

Beri et al. reported that their AI algorithm, trained to identify the motion-affected regions by segmenting the entire image series, had an AUC of 0.81 [9]. We achieved a similar AUC of 0.88 with a single coronal MPR image per CTPA examination. Other studies [10,23,24] on motion artifact detection in CT images have focused on coronary CT angiography (CCTA) rather than CTPA examinations. Ma et al. reported 91% sensitivity and 71% specificity with 87% accuracy for detecting motion artifacts in CCTA examinations [24]. Likewise, Elss et al. trained an AI algorithm for identifying motion artifacts on CCTA and reported an accuracy of 94% [10]. Xu et al. reported a fully automatic AI for grading image quality (motion artifacts) of CCTA using semi-automatic labeling and tracking of the coronary arteries [11]. Based on the identification and estimation of motion artifacts, other investigators have reported on motion artifact correction and compensation solutions for head and cardiac CT examinations [25,26]. 

Compared to the 94% sensitivity of the AI model in the training datasets, the 85% sensitivity in the test dataset suggests a drop in the performance of the AI model on the external dataset. This could be related to the differences in the prevalence and severity of motion artifacts or the CT image acquisition parameters between the test and training sites. The drop in sensitivity in the test datasets could also be related to an unknown confounding factor, such as the presence and distribution of lung abnormalities on chest CT exams from different sites. 

Although not yet cleared by the US Food and Drug Administration (FDA), our proof-of-concept study and the AI model used in the study may have potential clinical implications. Firstly, given the high frequency of motion artifacts in chest CT and CTPA examinations, our study highlights the role of AI models in identifying motion artifacts. If integrated with CT scanners, such AI models can efficiently detect and alert the CT technologist to motion artifacts likely to substantially affect the diagnostic interpretation. Secondly, such artifacts can be present on chest CT exams acquired on the older, less advanced scanners and newer, faster scanners. On the latter, the modification of scanning parameters can enable faster scanning of the entire chest in under 1 s [27]. The AI-generated surveillance for motion-impaired CTPA or chest CT examinations on the older scanner can prompt technologists to give better breath-hold instructions [12], modify scan parameters, or scan at-risk patients on faster scanners when possible. Thirdly, most diagnostic CT scanners in our institution (MGH) have two scan protocols—one for patients who can hold their breath and the other for those who cannot hold their breath or have substantial motion artifacts on their initial CT acquisition. Despite instructions to our CT technologists to always review the CT images for motion artifacts before taking the patient off the CT table, most technologists cannot or do not comply with the recommendation, especially during pandemic times. As a result, the interpreting radiologists either report CT with a disclaimer on motion-limited diagnostic value or request patient recall and rescanning. By automating the detection of motion artifacts, AI models such as the one reported in our study could potentially help address compliance and reacquisition when appropriate. Fourthly, several modern scanners automatically generate multiplanar reformatted images as soon as the data acquisition is complete, so using a coronal MPR image for the model is not a rate-limiting step. However, the model would still require the identification of the single image at the descending aorta level. Finally, in hospitals with high CT volumes, the quality assessment task for image quality is time consuming and labor intensive. In such sites, the AI model used in this study can analyze image quality retrospectively. The derived statistics can then be used to develop faster scan protocols and track their impact on diagnostic evaluability. Our study has limitations. We did not perform a power analysis to determine the number of training and testing cases needed to prove our hypothesis. Although the high level of performance associated with the AI model suggests that our sample size was adequate, it is conceivable that more training data could generate better results. Likewise, for external testing, we had only a single site with completely independent clinical operations, and we would require additional external sites to support the claim of the generalizability of the AI model. However, the AI model’s performance can vary with the change in scan protocols (low-dose chest CT versus CTPA protocols) and scanners (for those scanners without input training data). The exclusion of CTPA with “white lungs” (diffuse parenchymal opacities), substantial bilateral lung volume loss, or pneumonectomies also limits the application of the AI model in such patients.

Another limitation of our study pertains to using a single coronal MPR image per CT for assessing substantial motion artifacts as opposed to the entire image series for prior studies [9]. It is, therefore, possible that the performance of the AI model can differ on the entire image series (transverse or coronal) compared to its current performance on a single image. Despite a high model performance, it is possible that motion artifacts in other anatomic locations can affect the evaluation of key findings. Given the full longitudinal coverage of anatomy in the coronal plane, we believe that such “missed motion artifacts in key locations” are less likely. The AI building platform at the time of manuscript preparation cannot easily group entire image series and is limited to 2D image input, which impacted our decision to train on the coronal MPR image instead of an axial series. Finally, the model training and testing were limited to CTPA and might not apply to other chest CT protocols or body regions. 

## 5. Conclusions

The physician-trained and tested AI model can help identify substantial motion artifacts in CT pulmonary angiography. Automatic recognition of such artifacts can help CT technologists apply faster scan protocols and reacquire images to mitigate the impact of substantial motion impairment on diagnostic evaluability. 

## Figures and Tables

**Figure 1 diagnostics-13-00778-f001:**
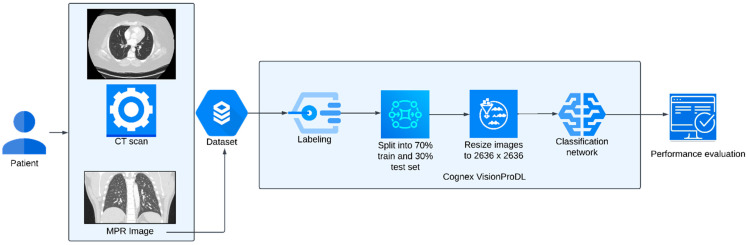
Diagrammatic representation of model architecture and training workflow.

**Figure 2 diagnostics-13-00778-f002:**
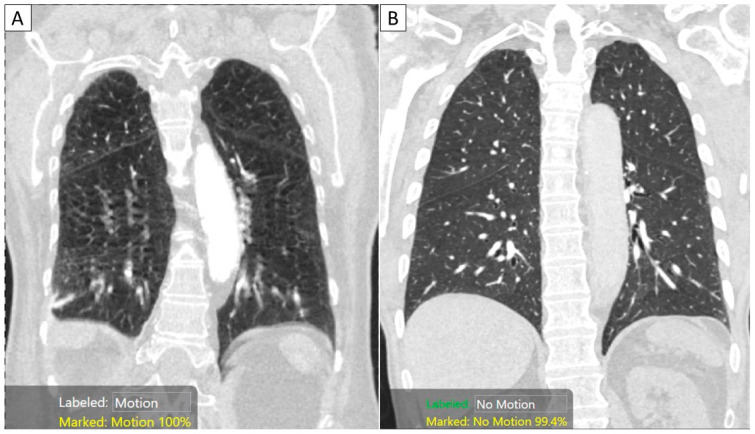
Coronal MPR images of CTPA examinations in two patients with (**A**) and without substantial motion artifacts. The AI algorithm correctly classified the images with motion ((**A**) with a 100% confidence score) and “no motion” ((**B**) with a 99.4% confidence score).

**Figure 3 diagnostics-13-00778-f003:**
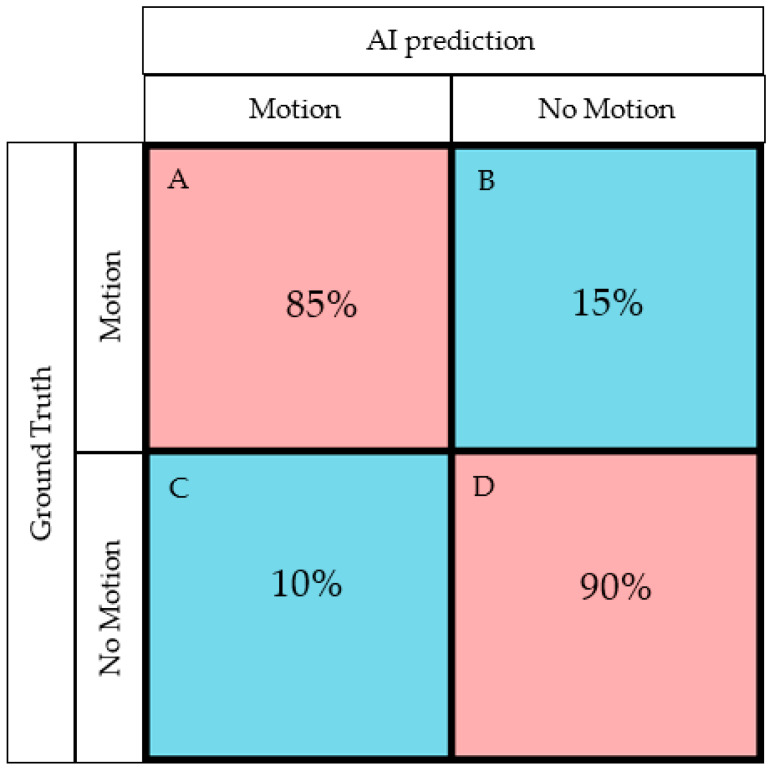
Confusion matrix generated by the AI model for the testing (A true positive rate (sensitivity)—AI correctly labeled images with motion; B false-negative rate—AI incorrectly labeled “motion” images as without motion; C false positive rate—AI incorrectly labeled “no motion” images as with substantial motion artifacts; D true negative rate (specificity)—AI correctly identified images without motion artifacts).

**Figure 4 diagnostics-13-00778-f004:**
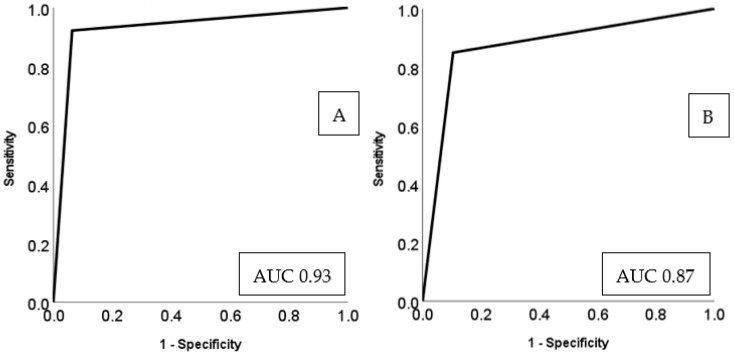
Area under the curve (AUC) for the performance of the AI model for validation (**A**) and external testing (**B**).

**Table 1 diagnostics-13-00778-t001:** Summary of the AI model’s performance for motion artifact detection in CTPA exams from the three participating sites (A, B, C). (key: +, present; -, absent; CI, confidence interval).

		Training	Test	Sensitivity	Specificity	Accuracy	AUC	95% CI (AUC)
Model validation (training + validation data from all sites)	Motion +	295	126	94%	91%	93%	0.93	0.89–0.97
	Motion -	259	113					
Model testing (training from A, C, and testing on B)	Motion +	196	173	85%	90%	86%	0.87	0.82–0.92
	Motion -	231	86					

## Data Availability

All data generated are reported within the article.

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
