# Peer review of "Auto-Detection of Motion Artifacts on CT Pulmonary Angiograms with a Physician-Trained AI Algorithm"

_diagnostics, 2023, doi:10.3390/diagnostics13040778_

Round 1
Reviewer 1 Report (Previous Reviewer 2)
I mean by the architecture of the model, what kind of model is used?
The paper seems nice, but still needs improvement:
- English editing.
-state of art is noyt incliudede.
Author Response
I mean by the architecture of the model, what kind of model is used?
The classification tool used is similar to classification neural networks such as VGG, ResNet and DenseNet. The segmentation tool is similar to the segmentation neural network, such as U-Net.
The paper seems nice, but still needs improvement:
- English editing.
Thank you for the suggestion. Two of the co-authors are native English speakers, who reviewed the paper for English editing. The changes have been made.
-state of art is noyt incliudede.
Thank you for the suggestion. To our best knowledge, only one reported literature on the identification of motion artifacts on Chest CT has been reported. This study, along with prior studies of motion artifact detection on coronary CT angiography, has been compared in the second paragraph of the discussion.
Reviewer 2 Report (New Reviewer)
Review Report:
Minor comments:
Please consider adding this important points and some recent literature such as:
https://doi.org/10.1007/s40571-022-00490-w
The manuscript is not well documented and slight modifications, it will be perfect.
Please improve images and take inspiration from recent works.
Major comments:
Ø
Please cite recent literature where these approaches are used and discussed in a more convincing manner:
https://doi.org/10.1016/j.sintl.2022.100202
Kindly improve the presentation of graphs and schematics.
I would suggest citing theory base works as above and then write in brackets (and the references therein). This will increase the authenticity of your findings.
Ø
The outcomes were really poorly documented.
I suggest presenting comparative analysis via images or table and properly add citations. For reference how to, please see:
https://doi.org/10.1016/j.sintl.2022.100202
Ø
When such medical images are processed, their data repository is properly cited as showed in mentioned works in my review. I would like to guide the authors to split acknowledgments to two parts, one where they will mention the funds, one, where they will mention the source of medical data, the source of codes.
Author Response
Minor comments:
Please consider adding this important points and some recent literature such as:
https://doi.org/10.1007/s40571-022-00490-w
The manuscript is not well documented and slight modifications, it will be perfect.
Please improve images and take inspiration from recent works.
- Both the references have been cited as explainable AI in the section on ‘Model’ in materials and methods.
Major comments:
Ø Please cite recent literature where these approaches are used and discussed in a more convincing manner:
https://doi.org/10.1016/j.sintl.2022.100202
Kindly improve the presentation of graphs and schematics.
I would suggest citing theory base works as above and then write in brackets (and the references therein). This will increase the authenticity of your findings.
- Thank you for the excellent suggestion and examples for the illustration. We have updated figure 2 with a similar representation of the confusion matrix.
Ø The outcomes were really poorly documented.
I suggest presenting comparative analysis via images or table and properly add citations. For reference how to, please see:
https://doi.org/10.1016/j.sintl.2022.100202
- Thank you for the suggestion. The analysis performed in the above-mentioned paper compares multiple variables in both the control and study group. However, our study involved only two groups: with and without motion artifacts.
Ø When such medical images are processed, their data repository is properly cited as showed in mentioned works in my review. I would like to guide the authors to split acknowledgments to two parts, one where they will mention the funds, one, where they will mention the source of medical data, the source of codes.
- The details are as follows:
Funding: There study was not supported by any funding.
Source of data and code: The data utilized in the study was acquired from one of the three US hospitals, as mentioned in the paper. We used a commercial product to train the AI model (COGNEX Inc.). Therefore, we do not have access to the source codes (https://support.cognex.com/en/downloads/visionpro-deep-learning).
Reviewer 3 Report (New Reviewer)
Dear authors
In this paper, “Auto-detection of motion artifacts on CT pulmonary angiograms with a physician-trained AI algorithm” is presented. This manuscript is not standardized and detailed enough. It is suggested to make a major revision before submitting it.I have some major concerns about this paper:
1.Manuscripts with track changes are very irregular.
2.The proposed model should be compared with the existing models.
3.References in the text are not standardized, some use "(", some use "[".
4.The references in the last part are not standardized and lack of time, publishing unit, page number, etc
5.There are many syntax errors. For example,
(1) in the same sentence pattern of the summary, one uses "belonged" and the other uses "belonging".
(2)"Report" should be replaced by "reported" in line 214 on page 7.
6.Both the figure 2 and figure 3 "ROC" plot are not standardized enough.
7.False negative is relatively high, and no explanation was given.
Author Response
Dear authors
In this paper, “Auto-detection of motion artifacts on CT pulmonary angiograms with a physician-trained AI algorithm” is presented. This manuscript is not standardized and detailed enough. It is suggested to make a major revision before submitting it.I have some major concerns about this paper:
1.Manuscripts with track changes are very irregular.
- We apologize for the irregularity. As multiple reviewers requested changes, the manuscript has many changes.
- The proposed model should be compared with the existing models.
- To our best knowledge there has been only one reported literature on identification of motion artifacts on Chest CT. This study along with prior studies of motion artifact detection on coronary CT angiography has been compared in the second paragraph of the discussion.
3.References in the text are not standardized, some use "(", some use "[".
- All the references have been standardized to “()”.
4.The references in the last part are not standardized and lack of time, publishing unit, page number, etc
- The references have been rechecked and corrected with the appropriate format.
5.There are many syntax errors. For example,
(1) in the same sentence pattern of the summary, one uses "belonged" and the other uses "belonging".
(2)"Report" should be replaced by "reported" in line 214 on page 7.
- The requested changes have been made. The paper has been reviewed by two co-authors who are native English speakers.
6.Both the figure 2 and figure 3 "ROC" plot are not standardized enough.
- Figure 2 has been changed to represent a standardized confusion matrix.
7.False negative is relatively high, and no explanation was given.
- We have added the explanation related to the high false positive rate in the discussion.
Round 2
Reviewer 3 Report (New Reviewer)
The author has basically revised it. I have no other suggestions.
This manuscript is a resubmission of an earlier submission. The following is a list of the peer review reports and author responses from that submission.
Round 1
Author Response
Thank you for the valuable suggestions. Please see the attachment.

Reviewer 2 Report
The paper I very well presented. Nonetheless, I can't really understand what kind of model train. deep learning model? architecture?
No state of art resume, hence the novelty of the paper is not clear. I suggest explaining the learned model architecture in detail so the reader will have a global idea of what you are introducing as potential new content.
Author Response
Thank you for the suggestions. The model is a deep learning model. We have added the information of the model and provided the prior study that provides further details as a reference.
Round 2
Reviewer 1 Report
The authors' are using Vision Pro Deep Learning software for motion artifact detection. Was the software developed by the authors? What was their AI model? Did the software come from a third party? It's not clear. They are saying "our AI model". There is no description of the deep learning model (architecture) and its parameters and model-related other stuffs. They have just mentioned the advantages of using the software without any programming knowledge. What was the authors' contribution in this regard?
Author Response
Dear Reviewer,
Thank you very much for the suggestions and comments. The software for the model development was developed by a third-party commercial vendor, COGNEX Corporation, Natick, MA, as mentioned on page 3, line 116. We used the platform to develop this AI model to identify the motion artifact.
Due to the proprietary rights of the vendor, the detailed model architecture cannot be provided due to the commercial interests of the vendor (personal communication with COGNEX, Michael MacDonald). However, we have added a new flowchart explaining the process of model development with specifications. The purpose of the study was to validate if non-programmers can use such model-building platforms to develop the models.
Reviewer 2 Report
The model needs to be explained with a schema.
I believe as well that you need to improve the quality of the paper more, before considering publication. In terms of organizations specifically.
Author Response
Dear Reviewer,
Thank you for the suggestions for improving the manuscript. Due to the proprietary rights of the vendor, the detailed model architecture cannot be provided due to the commercial interests of the vendor (personal communication with COGNEX, Michael MacDonald). However, we have added a new flowchart explaining the process of model development with specifications.
Our manuscript has been organized in accordance to CLAIM guidelines (reference 18), which provide checklist and organization guidelines for AI manuscripts in medical imaging.
Round 3
Reviewer 1 Report
The authors have used a platform developed by a third-party commercial vendor, COGNEX Corporation, Natick, MA. They have just validated the vendor's AI platform with a dataset for motion artifact detection. Is there any significance? You have mentioned that due to the proprietary rights of the vendor, the detailed model architecture cannot be provided due to the commercial interests of the vendor.